



Atmospheric
Measurement
Techniques

# Retrieval algorithm for the column $CO_2$ mixing ratio from pulsed multi-wavelength lidar measurements

**Xiaoli Sun**[1], **James B. Abshire**[1,2], **Anand Ramanathan**[1,a], **Stephan R. Kawa**[1], and **Jianping Mao**[1,2]

[1]NASA Goddard Space Flight Center, Science and Exploration Directorate, Greenbelt, Maryland, USA
[2]University of Maryland, College Park, Maryland, USA
[a]now at: Audible, Inc., Newark, New Jersey, USA

**Correspondence:** Xiaoli Sun (xiaoli.sun-1@nasa.gov)

**Abstract.** The retrieval algorithm for $CO_2$ column mixing ratio from measurements of a pulsed multi-wavelength integrated path differential absorption (IPDA) lidar is described. The lidar samples the shape of the 1572.33 nm $CO_2$ absorption line at multiple wavelengths. The algorithm uses a least-squares fit between the $CO_2$ line shape computed from a layered atmosphere model and that sampled by the lidar. In addition to the column-average $CO_2$ dry-air mole fraction ($XCO_2$), several other parameters are also solved simultaneously from the fit. These include the Doppler shift at the received laser signal wavelength, the product of the surface reflectivity and atmospheric transmission, and a linear trend in the lidar receiver's spectral response. The algorithm can also be used to solve for the average water vapor mixing ratio, which produces a secondary absorption in the wings of the $CO_2$ absorption line under humid conditions. The least-squares fit is linearized about the expected $XCO_2$ value, which allows the use of a standard linear least-squares fitting method and software tools. The standard deviation of the retrieved $XCO_2$ is obtained from the covariance matrix of the fit. The averaging kernel is also provided similarly to that used for passive trace-gas column measurements. Examples are presented of using the algorithm to retrieve $XCO_2$ from measurements of the NASA Goddard airborne $CO_2$ Sounder lidar that were made at constant altitude and during spiral-down profile maneuvers.

## 1 Introduction

Accurate remote sensing of atmospheric $CO_2$ from Earth-orbiting satellites is a key component in a long-term carbon–climate observing system (Sellers et al., 2018). Airborne and spaceborne lidar can be used to remotely monitor the global $CO_2$ and other trace-gas concentrations under conditions that are inaccessible to passive spaceborne $CO_2$ measurement missions, such as GOSAT (Kuze et al., 2016), OCO-2 (Crisp et al., 2017), and OCO-3 (Eldering et al., 2017, 2019). Studies have shown (Kawa et al., 2018) that a polar-orbiting integrated path differential absorption (IPDA) lidar can measure $XCO_2$ with low bias and high precision at all sun angles, seasons, and latitudes using a constant nadir-zenith illumination and observation geometry. A pulsed IPDA lidar also provides the range-resolved atmospheric backscatter profiles, so that return signals from the surface, clouds, and aerosols can be uniquely separated (Allan et al., 2019). This allows pulsed IPDA lidar to measure $XCO_2$ to the surface, cloud tops, or both (Ramanathan et al., 2015). Because the laser pulses reflected from clouds, aerosols, and surface are separated in time, the $XCO_2$ retrievals to the ground surface are not biased by scattering from clouds and aerosols (Mao et al., 2018).

Several types of dual-wavelength (online and offline) IPDA lidar have been demonstrated previously for measuring $XCO_2$ from aircraft (Spiers et al., 2011; Menzies et al., 2014; Jacob et al., 2019; Dobler et al., 2013; Campbell et al., 2020; Refaat et al., 2016, 2020, 2021; Amediek et al., 2017; Zhu et al., 2019, 2020). A multi-wavelength IPDA lidar has also been reported. The retrieval algorithms used in

these IPDA lidars calculate the ratios of online to offline atmosphere transmission, convert them to differential absorption optical depths (DAODs), and then solve for XCO$_2$ from the DAOD based on atmospheric transmission models. For the multi-wavelength lidar proposed by Han et al. (2020), a series of DAODs are calculated, and a least-squares fit is used to solve for XCO$_2$ (Han et al., 2020). The XCO$_2$ can be solved directly from the DAOD; however, these algorithms rely on the accurate knowledge of the line shape of the CO$_2$ absorption. They are sensitive to measurement biases due to uncertainties in spectroscopy and meteorological conditions that affect the line shape. They also require precise knowledge of the laser wavelengths, the lidar receiver optical transmission versus wavelength, and the Doppler shift of the received laser signals.

NASA Goddard Space Flight Center (GSFC) has developed an airborne multi-wavelength CO$_2$ sounder lidar and demonstrated XCO$_2$ measurements through a series of airborne campaigns (Abshire et al., 2010, 2013, 2014, 2018; Ramanathan et al., 2013, 2015, 2018; Mao et al., 2018). Its retrieval compares the lidar-sampled line shape with one computed from an atmosphere model to retrieve XCO$_2$. Several other parameters, such as the Doppler shift, surface reflectance, and lidar receiver spectral response are solved simultaneously via a least-squares fit. The retrieval algorithm is similar to those used for passive trace gas measurements with modifications specifically for the lidar measurement. Although this multi-wavelength approach requires more laser power to achieve a given XCO$_2$ measurement precision, it provides more tolerance to the uncertainties in the CO$_2$ absorption line shape, lidar receiver response, and Doppler shift, so that the retrieved XCO$_2$ is more robust against bias errors.

This paper describes the retrieval algorithm for the multi-wavelength CO$_2$ Sounder lidar and provides a framework for similar IPDA lidar for other atmospheric gas measurements. Parts of the algorithm have been reported earlier (Ramanathan et al., 2013, 2015, 2018). This paper gives a complete description of the algorithm, the mathematical derivations, signal processing techniques, estimation error, and averaging kernel. An example of using the algorithm to analyze measurements from the airborne CO$_2$ Sounder lidar is also presented.

## 2 Measurement approach

The measurement geometry for the CO$_2$ Sounder lidar is shown in Fig. 1. The lidar transmits laser pulses toward nadir, and its receiver telescope collects the optical signal backscattered from the atmosphere and the surface. Figure 2 shows a block diagram of the airborne CO$_2$ Sounder lidar, which was developed as an airborne demonstrator for NASA's planned Active Sensing of CO$_2$ Emissions over Nights, Days, & Seasons (ASCENDS) mission (Kawa et al., 2018). The laser

consists of a tunable seed laser, a pulsed modulator, and a power amplifier. The seed laser module consists of two single-frequency continuous-wave (CW) diode lasers. One is the reference laser whose wavelength is locked to the center of the CO$_2$ absorption line in a gas cell. The other diode laser (slave) is tunable and its wavelength is locked to that of the master, plus a programmable offset frequency. The offset frequency is step-tuned across the CO$_2$ absorption line. The number of laser wavelengths in the scan and the exact wavelength of each laser pulse are digitally pre-programmed and can be adjusted via software commands (Numata et al., 2012). An electro-optical modulator is used to gate the output of the slave laser into 1 µs wide pulses. The laser pulses are then amplified by a multi-stage commercial fiber laser amplifier. The airborne lidar's laser pulse rate is 10 kHz and there are 30 wavelengths per scan, which gives a wavelength scan rate of about 300 Hz. The transmitted laser pulse energy at each wavelength is also sampled, and the results are used to normalize the received signal to correct for fluctuations in the transmitted laser energy with wavelength.

The lidar receiver detects and records the received laser pulse waveform over the entire atmosphere column traveled by the laser pulses. In the airborne lidar all signals are digitized and recorded and the lidar analysis is performed on ground after the flight. The received signals from the scattering surface are used to retrieve XCO$_2$. The signals before the ground returns are used to obtain the atmosphere backscatter profiles as an ancillary data set. The signals recorded after the ground returns are used to estimate the solar background, the detector dark noise, and the baseline voltage offset in the detector output. The baseline offset is subtracted from the signal before calculating the ground return pulse energies. The times of flight of the laser pulse to the targeted scattering surface are used to determine the atmosphere column height over which the CO$_2$ is measured.

## 3 Lidar signal processing and atmosphere modeling

An overview of the retrieval algorithm for lidar data is shown in Fig. 3. The initial processing consists of (a) processing the stored lidar data to estimate ranges to the reflecting surfaces and form a series of atmosphere transmission measurements across the CO$_2$ absorption line; (b) generating a CO$_2$ absorption line shape from the radiative transfer model and meteorological data at the time and location of lidar measurements; and (c) performing a least-squares fit of the modeled line shape function to the measurements to solve for XCO$_2$ and other parameters.

### 3.1 Lidar signal processing

The signal waveforms are first corrected for the detector baseline offset and other instrument characteristics and then scaled to the received optical signal power. The pulse ener-

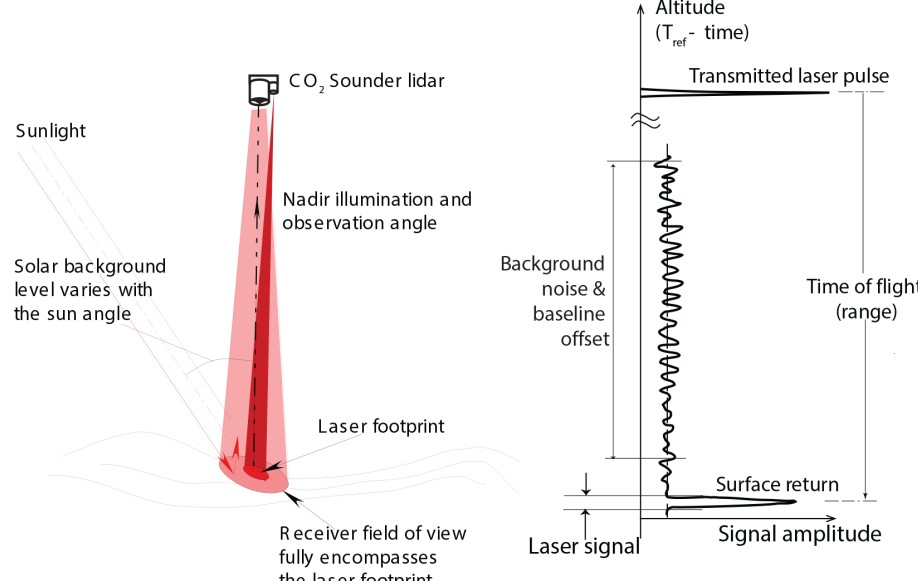

**Figure 1.** Illustrations of the CO$_2$ Sounder lidar measurement geometry and received signal at a single laser wavelength.

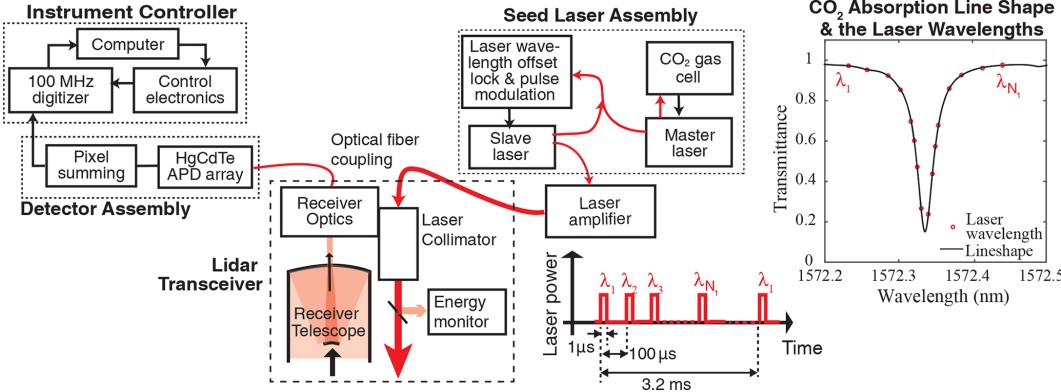

**Figure 2.** Block diagram of the airborne CO$_2$ Sounder lidar.

gies from the scattering surfaces are calculated by integrating the received pulse waveforms over the pulse width interval. The relative atmosphere transmittances for all laser wavelengths are calculated by dividing the received pulse energies by the transmitted ones and then multiplying by the square of the range from the lidar to the reflecting surface. The signal-to-noise ratio (SNR) of the atmospheric transmittances at each wavelength is estimated based on the received signal energy, the estimated background noise, and the detector noise. Finally, a least-squares fit of the modeled line shape to the lidar measurements is used to estimate XCO$_2$ along with the other parameters.

The lidar returns from clouds are identified by comparing the elevations of the lidar returns, namely aircraft altitude minus the lidar range, to the surface elevation from either the onboard radar measurements or a digital elevation model (DEM). For dense clouds, the laser energies reflected from the cloud tops are usually sufficient for XCO$_2$ retrievals (Mao et al., 2018). For thin clouds and aerosols, the laser pulses can often reach the ground surface and be received at the lidar with sufficient energy to allow useful XCO$_2$ retrievals. The signal waveform before ground return can be averaged to obtain the atmospheric backscatter profiles at the laser wavelength, which gives information about the heights and densities of thin clouds and aerosols (Allan et al., 2019).

### 3.2 Model for the lidar signals

The average signal pulse energy reflected from the scattering surface can be calculated from the lidar equation (McManamon, 2019), as

$$E_r(\lambda) = E_t(\lambda) \cdot T_A^2(\lambda) \frac{r_s}{\pi} \cdot \frac{A_r}{R^2} \cdot \eta_r, \qquad (1)$$

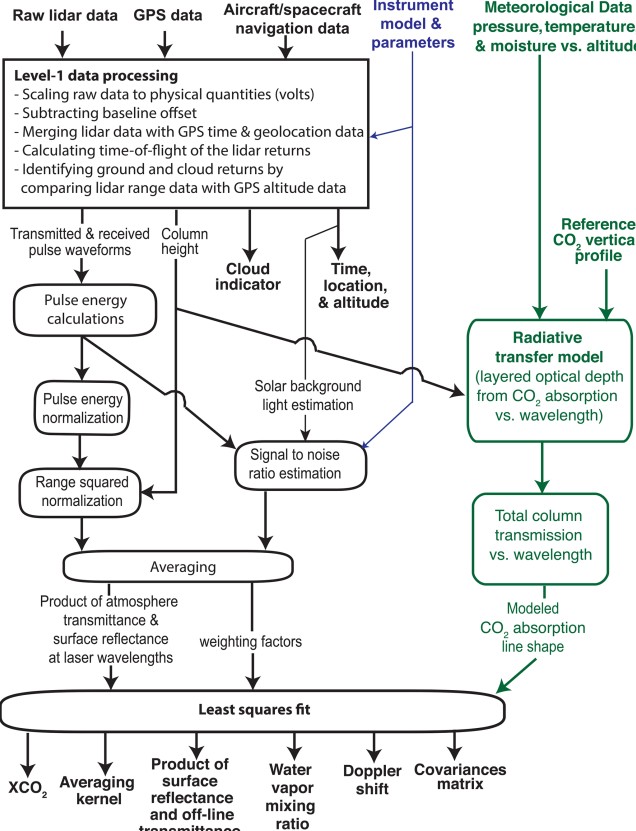

**Figure 3.** Flowchart of the XCO$_2$ retrieval algorithm for the CO$_2$ Sounder lidar.

where $E_r(\lambda)$ and $E_t(\lambda)$ are the received and transmitted laser pulse energies at laser wavelength $\lambda$, $T_A(\lambda)$ is the one-way atmosphere transmission at laser wavelength $\lambda$, $r_s$ is the diffuse surface reflectance to the laser beam, $A_r$ is the light collecting area of the receiver telescope, $R$ is the range from the lidar to the scattering surface (the column height), and $\eta_r$ is the receiver optical transmission efficiency.

The product of the surface reflectance and the two-way atmospheric transmission can be calculated from the received laser pulse energy after correcting for the range as

$$y(\lambda) = r_s T_A^2(\lambda) = \left(\frac{\pi}{A_r \eta_r}\right)\left[\frac{E_r(\lambda)}{E_t(\lambda)} \cdot \frac{1}{R^2}\right]. \quad (2)$$

The term in the parentheses of the right-hand side of Eq. (2) is a constant related to the lidar receiver. The telescope diameter and overall optical transmission are measured in the lab. The optical transmission can also be calibrated in flight by flying over an area where the surface reflectance and the atmospheric transmission are known from independent measurements. The term in brackets consists of variables measured by the lidar.

### 3.3 Model for the CO$_2$ absorption line shape

The total atmospheric transmission from the lidar to the surface can be written as

$$T_A^2(\lambda) = T_{CO_2}^2(\lambda) \cdot T_w^2(\lambda) \cdot T_o^2. \quad (3)$$

Here, $T_{CO_2}^2(\lambda)$ and $T_w^2(\lambda)$ are the two-way atmospheric transmissions of CO$_2$ and water vapor at laser wavelength $\lambda$. The term $T_o^2$ accounts for the transmission of aerosols and other particles, which are independent of the laser wavelength. $T_o^2$ is often called the offline atmospheric transmission. Note that Eq. (3) and the XCO$_2$ measurement are for the atmosphere column from the lidar to the surface. This is different from the passive remote sensing measurement where the incident light from the sun and the reflected light are at an angle and go through different atmosphere columns.

To compute the transmission line shapes of CO$_2$ and water vapor, the atmosphere is divided into a number of layers. The total transmission is modeled as the product of the individual transmissions of all the layers traveled by the laser pulse. The layered transmission is calculated from the layered radiative transfer atmospheric model, which takes into account the effects of the temperature, pressure, and humidity for each layer. The vertical profiles of temperature, pressure, and humidity are obtained from a meteorological analysis model or, when possible, from in situ atmospheric measurements made during aircraft spiral-down maneuvers.

The atmospheric transmissions of each layer for each of the lidar wavelengths across the CO$_2$ absorption line are calculated by using the Beer–Lambert law. The total two-way transmission due to CO$_2$ can be written as

$$T_{CO_2}^2(\lambda_i) = \exp\left[-2\sum_{j=1}^{N_2} \rho_{CO_2}(H_j)\,\sigma_{CO_2}(H_j, \lambda_i)\,\Delta H_j\right], \quad (4)$$

where $i = 1, 2, \ldots, N_1$ is the index for the laser wavelengths with $N_1$ the total number of laser wavelengths used in the lidar measurements, $j = 1, 2, \ldots, N_2$ is the index for the atmosphere layer with $N_2$ the total number atmospheric layers, $\rho_{CO_2}(H_j)$ is the molecular density of CO$_2$ for the $j$th layer, $H_j$ is the average altitude of the $j$th layer, and $\sigma_{CO_2}(H_j, \lambda_i)$ is the absorption cross section of a CO$_2$ molecule in the $j$th layer at the $i$th wavelength.

Here we assumed that the laser wavelengths are known precisely and the laser spectral line width is much narrower than the CO$_2$ absorption line width. For the CO$_2$ Sounder lidar, the laser is step locked to an onboard CO$_2$ gas cell with fixed frequency offsets in each scan. The frequency accuracy is $< 1$ MHz peak to peak, and the line width is about 30 MHz (Numata et al., 2012), which are small compared to the CO$_2$ absorption line width. It has been shown that for lidar measurements of XCO$_2$ such small laser frequency deviations are negligible compared to other noise sources (Chen et al., 2012, 2014, 2015, 2019).

The modeled optical transmission due to CO$_2$ can also be expressed in terms of the optical depth (OD) defined as the absolute value of the logarithm of the one-way atmospheric transmission, as

$$T_{CO_2}^2(\lambda_i) = \exp\left[-2OD_{CO_2}(\lambda_i)\right] \tag{5}$$

and

$$OD_{CO_2}(\lambda_i) = \sum_{j=1}^{N_2} \rho_{CO_2}(H_j)\sigma_{CO_2}(H_j,\lambda_i)\Delta H_j$$

$$= \sum_{j=1}^{N_2} \Delta OD_{CO_2}(H_j,\lambda_i), \tag{6}$$

where $OD_{CO_2}(\lambda_i)$ is the column OD at wavelength $\lambda_i$ and $\Delta OD_{CO_2}(H_j,\lambda_i) = \sum_{j=1}^{N_2}\rho_{CO_2}(H_j)\sigma_{CO_2}(H_j,\lambda_i)\Delta H_j$ is the OD of the atmosphere layer due to CO$_2$ absorption at wavelength $\lambda_i$ and altitude $H_j$.

The molecular density of CO$_2$ for the $j$th layer can be expressed as

$$\rho_{CO_2}(H_j) = XCO_2(H_j)\rho_{air}(H_j), \tag{7}$$

where $XCO_2(H_j)$ is the CO$_2$ mixing ratio and $\rho_{air}(H_j)$ is the dry-air molecular density at altitude $H_j$.

In our XCO$_2$ retrieval algorithm the layered OD is calculated by using the HITRAN 2008 spectroscopy database (Rothman et al., 2009) and the Line-By-Line Radiative Transfer Model (LBLRTM) V12.1 (Clough et al., 1992; Clough and Iacono, 1995), for a given CO$_2$ mixing ratio and meteorological vertical profiles at the time and location of the lidar measurement.

The atmospheric pressure, temperature, and water vapor can cause shifts and broadening of the CO$_2$ absorption line, which affects the cross sections at measured wavelengths. The LBLRTM software incorporates these effects and computes a numerical line shape function in OD at the given altitude of each layer. For the airborne data retrievals, meteorological data are obtained from the near-real-time forward processing of the Goddard Modelling and Assimilation Office (GMAO) FP system, the Goddard Earth Observing System Model, Version 5 (GEOS-5) (Rienecker et al., 2011). The data are drawn from the eight-per-day analyzed fields on the full model grid (0.25 × 0.3125° × 72 layers, inst3_3d_asm_Nv files). The GEOS-5 data are used for the meteorological conditions for the retrievals at the times and places where the airborne in situ profile measurements are not available. For analysis of our airborne campaign measurements, the GEOS-5 data were used primarily except during the spiral maneuvers. We extract the nearest-in-time latitude–longitude interpolated meteorological soundings from the GEOS-5 data every minute at regular positions along the flight's ground tracks. The 42 lowest analysis levels are used for each profile location. The analyzed pressure is used for the vertical grid coordinate for any of the profiles. The surface pressure and surface height are horizontally interpolated from the model.

Since the power and size of the CO$_2$ Sounder lidar are limited, there is a limit to the number of laser wavelengths which can be used to sample the XCO$_2$ absorption line at a given rate and maintain adequate SNR at each wavelength. Although Ramanathan et al. (2018) showed a few additional parameters about the CO$_2$ absorption line shape may be retrieved, they provide only limited information about the vertical profile of CO$_2$ mixing ratio. Therefore, we choose to retrieve a single scale factor for a reference profile, similar to the profile scaling used in passive remote sensing (Borsdorff et al., 2014). Here the reference profile is obtained from the radiative transfer model and meteorological data described above. A least-squares method is used to solve for the scale factor that minimizes the error between the line shape model and the lidar-sampled CO$_2$ absorption line shape at all laser wavelengths. This retrieval method assumes that the modeled line shape is accurate. In practice, there may be differences between the model and the actual line shape which could cause biases in the solutions. However, if the modeling error is random, the approach of using the lidar's sampling of the line at multiple wavelengths and using a line fit tends to average out the effect of the discrepancies.

Using the scale factor, the OD which is attributable to the CO$_2$ absorption can be written as

$$OD_{CO_2}(\lambda_i) \approx \alpha_{CO_2}\sum_{j=1}^{N_2} XCO_2a(H_j)\rho_{air}(H_j)$$

$$\times \sigma_{CO_2}(H_j,\lambda_i)\Delta H_j = \alpha_{CO_2}OD_{CO_2a}(\lambda_i), \tag{8}$$

where $\alpha_{CO_2}$ is the scale factor, $XCO_2a(H_j)$ is the a priori (initial guess) CO$_2$ mixing ratio at altitude $H_j$, and $OD_{CO_2a}(\lambda_i)$ is the a priori total column OD attributed to CO$_2$ absorption. The atmospheric transmission due to CO$_2$ absorption can now be approximated as

$$T_{CO_2}^2(\lambda_i) \approx \exp\left\{-2\alpha_{CO_2}OD_{CO_2a}(\lambda_i)\right\}. \tag{9}$$

## 4 Solving for XCO$_2$ from the lidar measurements via a least-squares fit

The column XCO$_2$ and several other variables are solved simultaneously from a least-squares fit of the modeled line shape to the lidar measurements. One variable is the Doppler shift in the wavelengths of the received signal, which occurs when measuring at non-nadir angles from a moving platform. Another parameter being solved for is the product of the surface reflectance and the two-way offline atmospheric transmission. For the CO$_2$ line at 1572.33 nm and under high humidity, there is a weak isotopic water vapor absorption feature on the left wing of the CO$_2$ absorption line. The retrieval algorithm can resolve this absorption feature to avoid causing biases in the retrieved XCO$_2$. For our airborne lidar, there

is also a small linear trend (slope) in the received laser pulse energy as a function of the wavelength. The primary cause of this trend is the residual error from modeling the uneven spectral response of the receiver optics, especially the optical bandpass filter. Since the bandpass spectral shape can change slightly with temperature and time, the retrieval also solves for this residual slope.

The least-squares fit may be formulated by expressing the lidar measurement data in matrix form, $\mathbf{Y}$, a single column matrix with elements $y_i$ given by Eq. (2). The parameter to be solved for, $\mathbf{S} = \{s_k\}$, is expressed as a $N_3 \times 1$ matrix. In our case $N_3 = 5$, where each element is defined as $s_1 = r_s T_o^2$ is the product of the surface reflectance and the two-way atmosphere transmission at offline wavelength, $s_2 = \alpha_{CO_2}$ is the scale factor for the XCO$_2$ line shape function, $s_3 = \alpha_{water}$ is the scale factor for the water vapor line shape function, $s_4$ is the linear slope of the receiver spectral response, and $s_5$ is the Doppler shift of the received signal wavelengths.

The modeled atmospheric transmission given in Eq. (9) can be expressed as a single column matrix, $\mathbf{F}(\mathbf{S})$, called a forward model, with each element equal to

$$
\begin{aligned}
f_i(\mathbf{S}) &= r_s T_A^2(\lambda_i, \mathbf{S}) \\
&\approx s_1 \left[ s_2 T_{CO_2}^2(\lambda_i + s_5) \right] \left[ s_3 T_{water}^2(\lambda_i + s_5) \right] \\
&\quad \times \eta_0(\lambda_i + s_5, s_4),
\end{aligned}
\tag{10}
$$

where $T_A^2(\lambda, \mathbf{S})$ is the atmosphere transmission defined in Eq. (1) but expressed as a function of both the laser wavelength and the parameters to be solved, and $\eta_0(\lambda, s_4)$ is the normalized receiver optical transmission as a function of the wavelength and the slope of the linear trend of the receiver spectral response. Here we also included the term for the water vapor.

A scalar-valued loss function can be defined as the sum of squared differences between the lidar measurement data and the model, as

$$
\begin{aligned}
J(\mathbf{Y}, \mathbf{S}) &= [\mathbf{Y} - \mathbf{F}(\mathbf{S})]^T \mathbf{W} [\mathbf{Y} - \mathbf{F}(\mathbf{S})] \\
&= \sum_{i=1}^{N_1} w_{i,i} \left[ y(\lambda_i) - f_i(\mathbf{S}) \right]^2,
\end{aligned}
\tag{11}
$$

where $[\mathbf{Y} - \mathbf{F}(\mathbf{S})]$ is an $N_1 \times 1$ matrix and $\mathbf{W}$ is a $N_1 \times N_1$ diagonal matrix for weighting factors used in the fit. The weighting factors are chosen to balance the contributions from the measurements at different laser wavelengths that have different SNRs. The least-squares fit finds the parameter set that minimizes the loss function.

For small changes in XCO$_2$ and for high SNR lidar measurements, Eq. (11) can be linearized by the first two terms of its power series expansion about initial estimates of the parameter values, $\mathbf{S0}$. The function $\mathbf{F}(\mathbf{S})$, also known as the forward model, can then be approximated by

$$
\mathbf{F}(\mathbf{S}) \approx \mathbf{F0} + \frac{\partial \mathbf{F}(\mathbf{S})}{\partial \mathbf{S}} |_{\mathbf{S}=\mathbf{S0}} (\mathbf{S} - \mathbf{S0}),
\tag{12}
$$

where

$$
\mathbf{F0} = \mathbf{F}(\mathbf{S0}) = \begin{bmatrix} f0(\lambda_1) \\ \vdots \\ f0(\lambda_{N_1}) \end{bmatrix}
$$

with $f(\lambda_i)$ equal to Eq. (10) evaluated at the initial value of the parameter set.

Substituting Eq. (12) into Eq. (11) and defining $\Delta \mathbf{Y} = (\mathbf{Y} - \mathbf{F0})$ and $\Delta \mathbf{S} = \mathbf{S} - \mathbf{S0}$, the loss function can now be approximated as

$$
\begin{aligned}
J(\mathbf{Y}, \mathbf{S}) &\approx \left[ \Delta \mathbf{Y} - \frac{\partial \mathbf{F}(\mathbf{S})}{\partial \mathbf{S}} |_{\mathbf{S}=\mathbf{S0}} \Delta \mathbf{S} \right]^T \\
&\quad \times \mathbf{W} \left[ \Delta \mathbf{Y} - \frac{\partial \mathbf{F}(\mathbf{S})}{\partial \mathbf{S}} |_{\mathbf{S}=\mathbf{S0}} \Delta \mathbf{S} \right].
\end{aligned}
\tag{13}
$$

For mathematical convenience, we normalize the lidar measurements with respect to their initial estimate and define a new variable:

$$
\Delta y1(\lambda_{i1}) = \frac{\Delta y(\lambda_i)}{f0(\lambda_i)}.
\tag{14}
$$

A diagonal matrix $\mathbf{I}_O$ can be defined with each element equal to $1/f0(\lambda_i)$. The loss function can be rewritten using the identity matrix $\mathbf{I} \equiv \mathbf{I}_O \mathbf{I}_O^{-1} \equiv \mathbf{I}_O^{-1} \mathbf{I}_O$, as

$$
\begin{aligned}
J(\mathbf{Y}, \mathbf{S}) &\approx \left[ \Delta \mathbf{Y} - \frac{\partial \mathbf{F}(\mathbf{S})}{\partial \mathbf{S}} |_{\mathbf{S}=\mathbf{S0}} \Delta \mathbf{S} \right]^T \left( \mathbf{I}_O \mathbf{I}_O^{-1} \right) \\
&\quad \times \mathbf{W} \left( \mathbf{I}_O^{-1} \mathbf{I}_O \right) \left[ \Delta \mathbf{Y} - \frac{\partial \mathbf{F}(\mathbf{S})}{\partial \mathbf{S}} |_{\mathbf{S}=\mathbf{S0}} \Delta \mathbf{S} \right] \\
&= \left[ \Delta \mathbf{Y1} - \frac{\partial \mathbf{F1}(\mathbf{S})}{\partial \mathbf{S}} \Delta \mathbf{S} \right]^T \\
&\quad \times \mathbf{W1} \left[ \Delta \mathbf{Y1} - \frac{\partial \mathbf{F1}(\mathbf{S})}{\partial \mathbf{S}} \Delta \mathbf{S} \right],
\end{aligned}
\tag{15}
$$

where $\Delta \mathbf{Y1} = \mathbf{I}_O \Delta \mathbf{Y}$, $\mathbf{F1}(\mathbf{S}) = \mathbf{I}_O \mathbf{F}(\mathbf{S})$, and $\mathbf{W1} = \mathbf{I}_O^{-1} \mathbf{W} \mathbf{I}_O$.

The use of the above normalization greatly simplifies the mathematical derivation as well as the data processing since it cancels out the exponential terms in the derivatives of $\mathbf{F}(\mathbf{S})$. However, this technique can only be used when the values of the forward model $f(\lambda_i)$ are not approaching zero at all sampling wavelengths.

The loss function given in Eq. (15) is of the same form as that of a linear least-squares fit with measurement data $\Delta \mathbf{Y1}$ and weighting factor $\mathbf{W1}$. The derivative of the function $\mathbf{F1}(\mathbf{S})$, which is often referred to as the Jacobian, is given by

$$
\mathbf{K} = \frac{\partial [\mathbf{F1}(\mathbf{S})]}{\partial \mathbf{S}} |_{\mathbf{S}=\mathbf{S0}}.
\tag{16}
$$

For the CO$_2$ Sounder lidar, each term of the Jacobian can be derived as $k_{i,1} = \frac{\partial \mathbf{F}(\mathbf{S})}{s_1} \frac{1}{f0(\lambda_i)} = \frac{1}{\langle r_s T_o^2 \rangle}$, same for all $i = 1, 2, \ldots, N_1$; $k_{i,2} = -2 OD_{CO_2 a}(\lambda_i)$, one for each laser wavelength, $i = 1, 2, \ldots, N_1$; $k_{i,3}$, same as above but for water

vapor; $k_{i,4} \approx \frac{T^2_{\mathrm{CO_2}}(\lambda_i + \Delta\lambda) - T^2_{\mathrm{CO_2}}(\lambda_i)}{\Delta\lambda} \cdot \frac{1}{\langle T^2_{\mathrm{CO_2}}(\lambda_i)\rangle}$ TS1, with $\Delta\lambda = 1\,\mathrm{pm}$ and $T^2_{\mathrm{CO_2}}(\lambda_i)$ given by Eq. (4); and $k_{i,5} = (\lambda_i - \lambda_c)$, with $\lambda_c$ the center wavelength of the CO$_2$ line shape function.

For measurement noise that is zero mean and follows a Gaussian distribution, the optimal weighting factors are given by the reciprocal of the variance of the measurement data (Bevington, 1969). In our case, the optimal weighting factors can be approximated as

$$w1_{i,i} = \frac{1}{\mathrm{var}\{\Delta y1(\lambda_i)\}} = \frac{f0(\lambda_i)^2}{\mathrm{var}\{y(\lambda_i)\}} \approx \frac{\langle y(\lambda_i)\rangle^2}{\mathrm{var}\{y(\lambda_i)\}}$$
$$= \mathrm{SNR}(\lambda_i)^2, \tag{17}$$

where $\langle y(\lambda_i)\rangle$ is the average value of the lidar measurement which is assumed to be close to the initial estimate $f0(\lambda_i)$. Therefore, for each wavelength the weighting factors can be approximated by the SNR of the lidar measurement at that wavelength. As mentioned earlier, the SNRs are calculated based on signal energy and background noise estimated from received pulse waveforms.

The XCO$_2$ and other parameters can now be solved using a standard linear least-squares fitting method with the loss function Eq. (15), Jacobian Eq. (16), and weighting factors Eq. (17). The solutions can be obtained numerically using the pseudo inverse function, as

$$\Delta\hat{\mathbf{S}} = \mathbf{G}\Delta\mathbf{Y1} \tag{18}$$

with $\mathbf{G}$ a $N_3 \times N_1$ matrix, which is often called the gain matrix and can be computed from the pseudo inverse function $\mathbf{pinv}(\cdot)$ (Peters and Wilkinson, 1970), as

$$\mathbf{G} = \mathbf{pinv}\left[\mathbf{K}^{\mathrm{T}}\mathbf{pinv}(\mathbf{W1})\mathbf{K}\right]\mathbf{K}^{\mathrm{T}}\mathbf{pinv}(\mathbf{W1}). \tag{19}$$

The pseudo inverse matrix function can be found in the MATLAB software package and in other software tools.

The covariance of the parameters can be obtained from Eq. (18), as

$$\mathbf{cov}\left(\Delta\hat{\mathbf{S}}\right) = \mathbf{G}\,\mathrm{var}\,(\Delta\mathbf{Y1})\,\mathbf{G}^{\mathrm{T}}, \tag{20}$$

with $\mathrm{var}(\Delta\mathbf{Y1})$ a diagonal matrix with each element the reciprocal of the corresponding element in Eq. (17). The covariance matrix $\mathbf{cov}(\Delta\hat{\mathbf{S}})$ is in general not a diagonal matrix even though $\mathrm{var}(\Delta\mathbf{Y1})$ is a diagonal matrix.

The variances of the estimated parameters are given by the diagonal elements of $\mathbf{cov}(\Delta\hat{\mathbf{S}})$. However, variance is only one of the criteria of the XCO$_2$ retrieval. There can still be a bias in the estimated parameters if there is a mismatch between the measurements and the modeled line shape.

The total column averaging kernel can be calculated as (Borsdorff et al., 2014)

$$\mathbf{A} = \mathbf{G}_{\boldsymbol{\alpha}}\mathbf{K}_{\mathbf{x}}, \tag{21}$$

where $\mathbf{G}_{\boldsymbol{\alpha}}$ is the row of the $\mathbf{G}$ matrix for calculating the XCO$_2$ scale factor and $\mathbf{K}_{\mathbf{x}}$ is the Jacobian of the measurement with respect to the layered CO$_2$ mixing ratios, which is an $N_1 \times N_2$ matrix given by

$$\mathbf{K}_{\mathbf{x}} = \frac{\partial\mathbf{F}(\mathrm{XCO_2})}{\partial\mathbf{XCO_2}}. \tag{22}$$

Each term of $\mathbf{K}_{\mathbf{x}}$ can be written according to Eqs. (2)–(4), (7), and (10), as

$$K_x(i,j) = \rho_{\mathrm{air}}(H_j)\,\sigma_{\mathrm{CO_2}}(H_j, \lambda_i)\,\Delta H_j.$$

The linear least-squares fit can also be iterated by correcting for the Doppler shift of the received laser wavelengths of the modeled line shape based on the solution from the previous iteration. The Jacobian terms are recalculated about the updated linearization point in each iteration to improve the results.

## 5 Evaluation of the retrieval algorithm using airborne lidar data

The algorithm described here was used to retrieve XCO$_2$ from measurements of our 2017 airborne lidar campaign (Mao et al., 2019). The lidar and the airborne measurements have been described in detail in Abshire et al. (2018). Table 1 lists the instrument parameters relevant to the XCO$_2$ retrieval. Here we show a few examples of using the retrieval algorithm on a data set collected during one of the 2017 flights. We also show the retrieved XCO$_2$ at different altitudes in comparison to XCO$_2$ calculated from the in situ measurements made during two spiral-down maneuvers.

Figure 4 shows an example of a Level-1 data set from our 2017 airborne campaign. It shows 30 transmitted pulse waveforms and the corresponding received pulse waveforms averaged over 32 laser wavelength scans. The decrease (tilt) of laser pulse amplitudes over the pulse width interval is caused by the depletion of energy stored in the laser gain media, which does not affect the IPDA lidar measurements. The energies of the transmitted laser pulses at different wavelengths fluctuate by a few percent, which is monitored and corrected for in the signal processing. The tails in the transmitted pulse waveforms shown in Fig. 4b are caused by an artifact of the laser monitor detector, which is different from the one used in the receiver. The amplitudes and energies of the received laser pulse waveform plotted in Fig. 4c clearly show the CO$_2$ absorption near the center of the wavelength scan. The XCO$_2$ retrieval is carried out at 1 Hz, during which the host aircraft typically travels about 200 m.

For the least-squares fit the weighting factor for each wavelength is the square of the SNR of the lidar-detected signals at that wavelength. The SNRs are estimated from the received lidar signal as (Gagliardi and Karp, 1995)

**Table 1.** The airborne CO$_2$ Sounder lidar instrument parameters.

| Instrument parameters | Values |
|---|---|
| **Laser** | |
| Pulse energy | 25 µJ |
| Wavelength scan range | 1572.235–1572.440 nm |
| Number of wavelengths | 30 (see Fig. 5) |
| Wavelength accuracy | 0.008 pm (1 MHz) |
| Spectral line width | < 0.247 pm (30 MHz) |
| Pulse width | 1 µs |
| Pulse rate | 10 kHz |
| Divergence angle | 0.43 mrad (4.3 m laser spot size on ground from a 10 km altitude) |
| **Receiver optics** | |
| Telescope size | 20 cm diameter |
| Field of view | 0.50 mrad |
| Optical filter bandwidth | 1.4 nm |
| Total optical transmission | 81.3% |
| **Detector and receiver electronics** | |
| Quantum efficiency, including fill factor | 69 % |
| Responsivity | $6.39 \times 10^8$ V/W |
| Noise equivalent power (NEP), 16 pixels combined | 6.9 fW Hz$^{-1/2}$ |
| Signal sample rate | 100 MHz, 16 bits |
| Integration time for each XCO$_2$ retrieval | 1 s |
| Data recording duty cycle | 90 % |

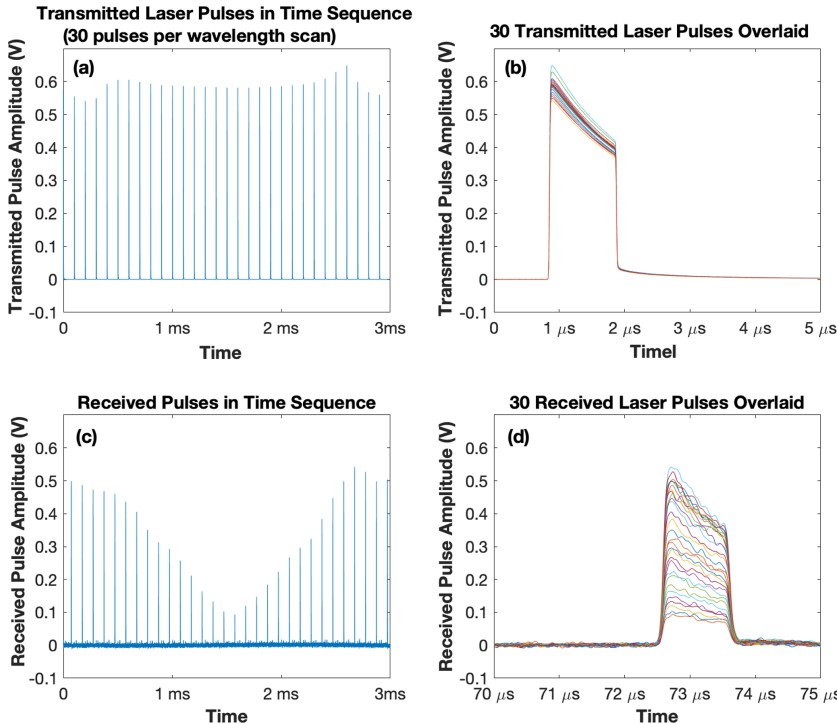

**Figure 4.** Sample pulse waveforms of the airborne CO$_2$ Sounder lidar taken during the flight of 21 July 2017 00:30:00 UTC (local time 20 July 2017 17:30:00). **(a)** The 30 transmitted laser pulse waveforms in a wavelength scan averaged over 32 repeated scans. **(b)** Overlay of the 30 transmitted pulse waveforms. **(c)** The corresponding received laser pulse waveforms reflected from the ground surface showing the lidar sampled CO$_2$ absorption. **(d)** An overlay of the received pulse waveforms for all 30 laser wavelengths.

$$\text{SNR}(\lambda_i) =$$

$$\frac{\eta_{\text{d}} \langle n_{\text{s}}(\lambda_i) \rangle}{\sqrt{F_{\text{d}} \left[ \eta_{\text{d}} \left( \langle n_{\text{s}}(\lambda_i) \rangle + \left(1 + \frac{\tau_{\text{s}}}{\tau_{\text{b}}}\right) \langle n_{\text{b}} \rangle \right) + \left(1 + \frac{\tau_{\text{s}}}{\tau_{\text{b}}}\right) \langle n_{\text{d}} \rangle \right] + \left(1 + \frac{\tau_{\text{s}}}{\tau_{\text{b}}}\right) \frac{\langle n_{\text{a}} \rangle^2}{\langle G_{\text{d}} \rangle^2}}}. \quad (23)$$

Here $\langle n_{\text{s}}(\lambda_i) \rangle$ is the average number of received signal photons per pulse at wavelength $\lambda_i$, $\langle G_{\text{d}} \rangle$ is the average gain of the avalanche photodiode (APD) detector, $\eta_{\text{d}}$ is the APD quantum efficiency, $F_{\text{d}}$ is the APD gain excess noise factor, $\tau_{\text{s}}$ is the integration time for the signal pulse, $\tau_{\text{b}}$ is the integration time for the background and dark noise, $\langle n_{\text{b}} \rangle$ and $\langle n_{\text{d}} \rangle$ are the average number of background photons and detector dark counts integrated over the pulse interval, and $\langle n_{\text{a}} \rangle$ is the standard deviation of the preamplifier noise in terms of equivalent number of photoelectrons. The signal here refers to the number of detected signal photons, which is equal to the number of the detected photons minus the number of detected background photons.

The laser speckle noise term (Goodman, 1965, 1975) is not included in Eq. (23) since it is not a major noise source for our airborne lidar measurements at nominal flight altitude. This is because of the large number of speckle cells in the laser footprint and the numerical averaging of the 30 received laser pulses for each XCO$_2$ retrieval. Laser speckle noise is also not expected to be a major noise source for the space version of the CO$_2$ Sounder lidar being developed at NASA GSFC (see chap. 5 of Kawa et al., 2018) since the effects of spatial and numerical averaging are similar. The effects of errors in the meteorological data used to construct the line shape model of CO$_2$ are also not considered in Eq. (23). We are currently conducting computer simulations to quantify the effect of meteorological data errors, and the results will be reported in a separate publication.

For the XCO$_2$ retrieval, the average number of received signal photons is estimated from the received pulse waveform. This is obtained by first integrating the received pulse waveform from the detector in volts, dividing the result by the detector responsivity in volts per watt, and the photon energy in joules. The average number of background noise photons is estimated from the average surface reflectance, offline atmosphere transmission, nominal sunlight irradiance on the surface, and receiver optics model. All other parameter values in Eq. (23) are instrument related and can be found in Abshire et al. (2018).

Figure 5 shows the CO$_2$ absorption line shape sampled by the lidar along with that from the forward model which assumes a constant XCO$_2$ vertical profile of 400 parts per million (ppm). It also shows the placement of laser wavelengths across the CO$_2$ absorption line. One laser wavelength (the second from the left) was placed at a secondary absorption feature due to deuterated water vapor (HDO). Three wavelengths were placed on the wings of the CO$_2$ absorption line. The rest were roughly equally spaced in OD along the absorption line. The residual differences between the measure-

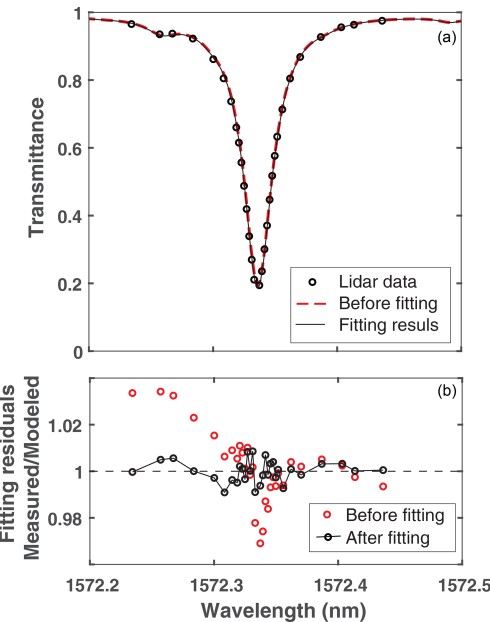

**Figure 5. (a)** An example of a CO$_2$ absorption line shape sampled by the airborne lidar (black circles) for 1 s averaging time and the models before and after the fit (red and black lines). **(b)** Differences (the residuals) between the lidar measurement and the model at the lidar wavelengths before (in red circles) and after (in black circles) the least-squares fit for the data set shown in Fig. 4. The retrieved scale factor was $\alpha_{\text{XCO}_2} \approx 1.025$ (i.e., 410 ppm retrieved vs. 400 ppm assumed).

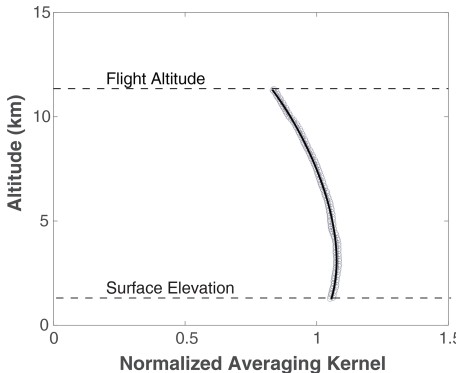

**Figure 6.** The averaging kernel from the retrieved data (open circles) and the fourth-order polynomial fit with altitude (solid black curve) from the measurement data shown in Fig. 5.

ments and the model after the least-squares fit are also plotted in Fig. 5. The averaging kernel is calculated based on Eq. (21) for each fit of 1 s lidar measurement data. Figure 6 shows the normalized averaging kernel with respect to its average value over the atmosphere column height and a fourth-order polynomial fit for the data shown above.

Figure 7 shows the results of the retrieval using the algorithm described above from the airborne CO$_2$ Sounder lidar measurements made on 21 July 2017 starting at

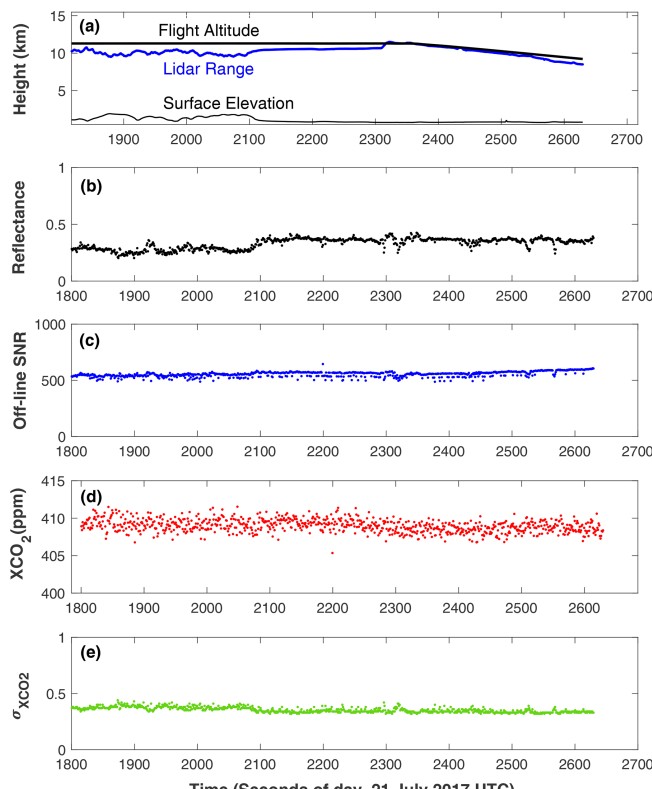

**Figure 7.** The results of the retrieval sequence from the airborne $CO_2$ Sounder lidar data starting at 21 July 2020 00:30:00 UTC for 820 s over Edwards Air Force Base in California. These are all based on 1 s receiver integration time. They are, from top down, **(a)** the aircraft altitude, the lidar range from the laser pulse time of flight, the surface elevation computed from the onboard GPS receiver, and the lidar range; **(b)** the retrieved surface reflectance times the two-way offline atmospheric transmission; **(c)** offline SNR calculated from Eq. (23); **(d)** the retrieved $XCO_2$; and **(e)** standard deviation of the retrieved $XCO_2$ from the covariance matrix. The airplane velocity was about 200 m s$^{-1}$. The distance covered by the data shown in the plot is about 164 km.

00:30:00 UTC for 820 s. The data consist of about a 500 s segment measured at a nearly constant aircraft altitude followed by about 300 s of measurements in a spiral descent. The last part of the flight was near Edwards Air Force Base, CA, and the surface elevation was nearly constant for the last 500 s. The ground surface over this stretch of the flight was dry desert, and the sky was visually clear at the time. The retrieved $XCO_2$ over this period was steady with a slow downward trend. The root-mean-squared (rms) variation in the retrieved $XCO_2$ from 2100 to 2300 s was 0.67 ppm, which includes both the fitting error and the actual $XCO_2$ variation along the flight path. By comparison, the estimated standard deviation from the retrieval covariance matrix was about 0.35 ppm, as shown in Fig. 7e.

Figure 8 shows the retrieved $XCO_2$ compared to that calculated from in situ measurements as the airplane flew in a

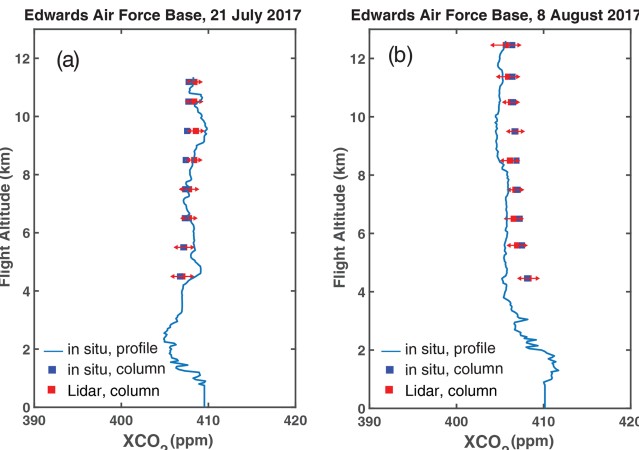

**Figure 8.** The column $XCO_2$ from the aircraft to the surface retrieved from the lidar measurements (red squares) compared to that computed from the in situ measurements (blue squares) during the spiral-downs on **(a)** 21 July 2017, near the end of the data segment shown in Fig. 7 and **(b)** 8 August 2017. The error bars on the red squares represent mean and the standard deviation of the lidar measurement results binned into 1 km layers. The blue squares are the column mixing ratio integrated from the readings of the AVOCET gas analyzer from the flight altitude to the surface. For the spiral-down comparisons, the median differences (biases) between the retrieved lidar and column in situ measurements are 0.72 ppm for 21 July and 0.16 ppm for 8 August 2017.

spiral-down path from 12 to 4 km over Edwards Air Force Base for the flight on 21 July 2017 and for that on 8 August 2017. The in situ profiles were measured by an updated version of the AVOCET gas analyzer on board the airplane (Vay et al., 2011). The in situ $XCO_2$ is calculated from the airplane altitude to the ground and is obtained by integrating the $CO_2$ profile from the in situ measurements weighted by the lidar's averaging kernel. The $XCO_2$ retrieved from the lidar measurements agrees with that calculated from the in situ measurements at all airplane altitudes above 4 km. Below 4 km, the laser beam no longer completely overlaps the field of view of the receiver, and the total $CO_2$ absorption (line depth) becomes small. The lidar measurements are not calibrated at such a low altitude.

# 6 Discussion

## 6.1 Biases in the retrieved $XCO_2$

Although the least-squares-fit method minimizes the sum of squared errors between the modeled line shape and the lidar measurements, it does not guarantee minimum biases in the estimated parameters. The variance of the solutions can approach zero as the SNR increases, as shown in Eq. (18), but biases remain. For example, if the actual $CO_2$ absorption line shape does not match that of the model, the retrieved

results can be biased regardless of the SNR. Therefore, it is important to model the atmosphere and the absorption spectroscopy accurately and avoid systematic errors.

## 6.2 Choice of laser wavelengths

The choice of the lidar laser wavelengths is a trade-off among several factors. The total number of laser wavelengths has to be greater than the number of parameters to be solved for in the retrieval; however, the total average laser output power is fixed. Using fewer wavelength samples allows improvement of the SNR for each sample but provides fewer constraints to the curve fit. More wavelength samples lower the SNR at each wavelength but allow us to solving for more parameters and helps to reduce the bias in the XCO$_2$ retrieval. There is also an advantage to select the laser wavelengths to be symmetrically distributed about the line center since it reduces the effect of the nonuniformity in the receiver's spectral response (Chen et al., 2019). Finally, the laser wavelengths should not be placed where the CO$_2$ absorption is too high, e.g., OD > 1.5, since the received signal level becomes too low to contribute to the retrieval.

The airborne CO$_2$ Sounder lidar mostly used 30 wavelengths with four offline, four near the center of the peak absorption up to OD = 1.2, one on the water vapor peak absorption, and the rest approximately uniformly distributed in OD (Abshire et al., 2018). This choice of the laser wavelengths produced measurement precisions < 1 ppm and biases < 1 ppm. Abshire et al. (2018) also report airborne measurements made using 15 laser wavelengths that showed no apparent difference in the XCO$_2$ measurements to those using 30 wavelengths for the otherwise same instrument configuration.

The retrieval algorithm described in this paper could also be used for the online and offline dual-wavelength IPDA lidar to retrieve XCO$_2$ and the product of surface reflectance and two-way atmosphere transmission. The solution to the least-squares fit for the two parameters can be derived analytically and becomes the same as those reported earlier (Abshire et al., 2010). The standard deviation of the retrieved XCO$_2$ at a given average laser power can be lower compared to that of a multi-wavelength IPDA lidar, depending on the placement of the online wavelength. However, the Doppler shift, water vapor content, and the receiver spectral response would have to be obtained and corrected well enough to avoid XCO$_2$ bias. The results would be much more sensitive to uncertainties in the CO$_2$ absorption line shape.

## 6.3 Number of parameters to retrieve

It is possible to use the least-squares fit to solve for more parameters of the CO$_2$ absorption line and lidar instrument, as long as the information content of the lidar measurements supports them. However, solving for more parameters, especially when they are correlated, increases the variance in the retrieved values, which limits the benefit. One example is to divide the atmosphere into a few layers, each with its own line shape function and scale factor, to obtain some information about the vertical distribution of XCO$_2$. The results from the least-squares fit for the XCO$_2$ for the layers, however, are correlated, and the errors from the fits are usually too large to be useful (Chen et al., 2014). A singular value decomposition (SVD) method has also been used to extract a few more parameters about the line shape without the need for an a priori vertical XCO$_2$ profile (Ramanathan et al., 2018). For measurement with high SNR, the SVD method can retrieve some characteristics of the line shape, such as the line width, and provide some constraints about the vertical distribution of XCO$_2$.

## 7 Conclusion

An algorithm to retrieve XCO$_2$ has been developed for measurements from a pulsed multi-wavelength IPDA lidar. The retrieval algorithm uses a least-squares fit of the line shape function derived from a multi-layer atmosphere radiative transfer model based on meteorological data to the line shape sampled by the lidar measurements. In addition to XCO$_2$, the algorithm simultaneously solves for the product of the surface reflectance and the offline atmosphere transmission, Doppler shift of the received laser signals, a secondary water vapor mixing ratio (if present), and a linear trend of the lidar receiver non-uniformity in its spectral response. Since it can accurately retrieve XCO$_2$ as these conditions vary, this approach provides a more robust measurement of XCO$_2$ compared to IPDA lidar that uses only online and offline wavelengths. The retrieval algorithm has been used successfully in the data processing of the NASA GSFC multi-wavelength pulsed IPDA lidar from its 2016 and 2017 airborne campaigns. The algorithm may also be used for retrievals for multi-wavelength lidars that target other atmospheric gases, such as CH$_4$.

*Code availability.* An IDL (Interactive Data Language) version of the software code for the least-squares fit will be posted at the same website by 1 July 2021 or contact the author xiaoli.sun-1@nasa.gov.

*Data availability.* The retrieved XCO$_2$ from the 2017 airborne lidar measurements is available from the NASA Airborne Science Data for Atmospheric Composition website, https://www-air.larc.nasa.gov/cgi-bin/ArcView/ascends.2017 TS2 (last access: 16 May 2021).

*Author contributions.* XS led the writing of the manuscript and provided the mathematical formulation of the retrieval algorithm. JBA was the principal investigator of the CO$_2$ Sounder lidar development and led the 2016 and 2017 ASCENDS airborne campaigns.

AR developed the retrieval algorithm and the data processing software. SRK and JM developed the atmospheric model used in the least-squares fit for the airborne measurement data processing. JM also processed and analyzed the 2017 airborne measurement data.

*Competing interests.* The authors declare that they have no conflict of interest.

*Acknowledgements.* We thank the $CO_2$ Sounder lidar team at NASA GSFC for the development of the lidar, conducting the airborne campaigns, and collecting the measurement data. We also thank Joshua P. Digangi for the AVOCET measurements, Julie Nicely for updating and testing the software for the $XCO_2$ retrieval, and Jeffrey Chen for many technical discussions about $XCO_2$ retrievals.

*Financial support.* This research has been supported by the NASA Earth Sciences Technology Office (ESTO) and the NASA AS-CENDS Mission pre-formulation program.

*Review statement.* This paper was edited by Markus Rapp and reviewed by two anonymous referees.

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

**Remarks from the typesetter**

TS1 Please provide a detailed explanation for those changes that can be forwarded to the editor. We will then start a process called post-review adjustments; once the editor approved the changes, we will insert them into the text.

TS2 Please provide a full reference list entry including creators and title (for further information, please see https://www. atmospheric-measurement-techniques.net/policies/data_policy.html).