# Peer review of "Retrieval Algorithm for the Column CO2 Mixing Ratio from Pulsed Multi-Wavelength IPDA Lidar Measurements"

_Atmospheric Measurement Techniques, 2020_

## Short Comment (SC1) · 31 Dec 2020

Regarding your reference to the work at NASA Langley (Dobler et al.), there is a more recent update to that you might want to reference also since there is a very large database of XCO2 measurements associated with it that is part of the Act America project lead by Ken Davis. This paper also outlines the algorithms used in computing XCO2. The paper is:

Field Evaluation of Column CO2 Retrievals From Intensity-Modulated Continuous-Wave Differential Absorption Lidar Measurements During the ACT-America Campaign

[Figure]

Joel F. Campbell, Bing Lin, Jeremy Dobler, Sandip Pal, Kenneth Davis, Michael D. Obland, Wayne Erxleben, Doug McGregor, Chris O'Dell, Emily Bell, Brad Weir, Tai-Fang Fan, Susan Kooi, Iouli Gordon, Abigail Corbett, Roman Kochanov

Earth and Space Science Volume7, Issue 12, December 2020 https://doi.org/10.1029/2019EA000847

There is also a database at:

https://daac.ornl.gov/cgi-bin/dataset_lister.pl?p=37
* * *

---

## Author Comment (AC1) · 31 Dec 2020

Thank you so much for the new reference. We will definitely include that in the updated manuscript. Happy New Year! Xiaoli Sun

---

## Referee Comment (RC1) · Anonymous Referee #1 · 19 Jan 2021

**General comments**

This is an interesting manuscript that addresses a novel approach to retrieve the column-integrated mixing ratio of greenhouse gases (XCO2 or XCH4) by measurements with a multi-wavelength pulsed Lidar sounder. XCO2, its variance and other parameters are derived from comparison of the measured transmission line shape function to a modelled one by means of a least squares fit. The presented retrieval method appears to be quite similar to passive sensors that but it is adjusted to serve for the pulsed Lidar measurements with its specific averaging kernels. For evaluation purposes, real multi-wavelength Lidar measurements by a specific airborne CO2 Sounder Lidar are used to test the proposed algorithm. The retrieved XCO2 values agree pretty well (< 1 ppm) to independent measurements of the mixing ratio column using aircraft in-situ data which is a remarkable result of this study. In a wider context the results of this study demonstrate that active remote sensing in combination to advanced retrieval methods fulfils the stringent measurement requirement on XCO2 for flux inversion experiments as demanded by the top down approach. The overall quality of the manuscript is good and the material presented is very well suited for publication in AMT subject to a few minor questions and comments for improvement of the manuscript as detailed below.

**Specific comments**

Chapter 1 lines 37-46: a brief chapter on the background and motivation for the selected measurement approach and retrieval algorithm is missing here. Different to passive sensors, conventional DAOD-based retrieval techniques don`t need a first guess CO2 profile for the calculation of XCO2. What is the reason to give-up this advantage or what is the benefit of the selected approach.

Further to this, the measurement precision of active sensors always lacks on sufficient "good" photons, particularly, if applied from space. The highest measurement sensitivity can be achieved at wavelength positions where the DAOD is close to unity (dependent on the background noise). This would imply to spend most of the laser pulses at wavelength positions that are more favorably in terms of measurement sensitivity rather than to emit in the far wing where the DAOD is very small as shown in Fig 2. Could the authors comment or justify the choice of the wavelengths. This question is also related to the discussion chapter 7.2

Chapter 2: Although the instrument description is not the focus of this issue a table of the most important instrument parameters which largely dominate the measurement performance from instrument point of view would be very beneficial for the manuscript. The missing parameters comprise: the laser pulse energy, the spectral width and spectral purity of the laser pulses, the wavelength domain of the pulse train, the optical filter band width, the laser footprint on ground, the off-nadir sounding angle, and maybe others. This table should also give information on the wavelength stability during a typical averaging period. In particular, the accuracy (stability) of the emitted laser pulses within each pulse train and its reproducibility over the various pulse trains during the measurement. The wavelength stability is a key parameter that impacts on the systematic error.

Chapter 3
Line 115: The transition from Eq: 1 to Eq: 2 is somewhat puzzling. More information on the calibration parameter C2 should be given. It is assumed that the calibration requirement comes from the energy monitor device which is different to the signal detection system in Fig.2. What is calibration procedure of this parameter on ground and what is the stability during airborne measurements.

Line 130:  The simple Eq. 4 assumes monochromatic laser pulses and almost zero emissions elsewhere (high spectral purity), otherwise it would fail. These important instrumental constraints should be added in this paragraph.

Lines 143-144: The introduction of the layered OD versus column OD is a bit confusing. What is measured and what is modelled? An equation for the measured column OD should be given here.

Line 150:  the line shift is also influenced by atmospheric parameters …   should be added.

Lines 161-163: It is agreed that the information content from measurement in the line wing appears to be rather small because of the small column OD in case of the selected CO2 absorption line. However, passive sensors that are capable to resolve the absorption line profile would face similar problems since a higher spectral resolution results in a lower SNR.  This is a confusing side discussion here, therefore it is recommended to drop these lines.

Lines 165-169:  Selection of only one scale factor might be a bit too optimistic. It would imply that the selected line parameters and the absorption cross section model are accurate enough to serve for all wavelengths and all atmospheric layers. Could the authors comment on this issue.

Line 210, Eq. 9:  further to above, the weighting factors introduced in the loss function are chosen to be similar at all wavelengths, just differing in their SNR. Maybe this is also a too simple approach and need to be justified in the manuscript. It takes not into account the various unfavorable column optical depths of the measurements in the line wing. The more optimal soundings with more optimal column ODs should be given more weight in the retrieval.

Chapter 5
Line 257:  What is the reason present Eq. 19.  It is not used in the manuscript

Chapter 6
Line 291:  What are other possible noise contributions such laser speckles in the Lidar echoes or from pulse energy detection unit which are not considered in Eq. 24.

Page 13 Fig. 5: Several parameters have been fitted in Fig. 5 but only the rms variation of XCO2 is discussed. What are the rms variations of the other fitted parameters and is there any interference to the retrieved XCO2.  Further to this, is the quality of the fit robust enough against changes of the first guess profiles CO2 and H2O or against errors of the spectroscopic or meteorological data.

Chapter 7.1
As known from many previous studies, systematic measurement errors play a key role in flux inversion experiments, and the requirements are very stringent there. The discussion on this topic is a bit thin. The authors are requested to discuss possible sources of systematic errors which may be related to the processing algorithms (linearization step, weighting factors, scale factor, first guess profiles, and spectroscopy to model the absorption line profile).

---

## Referee Comment (RC2) · Anonymous Referee #2 · 29 Jan 2021

The paper by Sun et al. discusses measurements of an airborne CO2 lidar that operates at multiple wavelengths, thus sampling the atmospheric transmission spectra of a single CO2 line. The paper puts much weight on the description of the least-squares retrieval algorithm for the (below-aircraft) column-average dry-air mole fraction of CO2 (XCO2) and the retrieval diagnostics. The topic is suitable for AMT and the demonstrator results in section 6 show excellent performance. The paper is a bit heavy on the algorithm description (which actually is a standard technique) but this might be justified since the paper aims at making a link from the lidar community to the community that works on passive remote sensing of CO2. Overall, I recommend publication after considering my mostly minor comments below.

Comments

1. The least-squares fit of an atmospheric transmission spectrum is not new. Its detailed description in section 2 through 5 might be justified in the present context where a new experimental technique is further developed. I would see one purpose in connecting the CO2 lidar community to the passive remote sensing community. But, then, the notation chosen by Sun et al. does not really correspond to standard notation e.g. defined in Rodgers, 2000, https://doi.org/10.1142/3171. I find it particularly twisting to use various "overhead" symbols for discriminating between true, a priori and estimated states. While it is not a mandatory request, I would recommend considering to go closer to standard notation.

2. I was particularly puzzled by the section on averaging kernels. It is a well-known problem that total-column/profile-scaling retrievals complicate the averaging kernel calculation. The most straight-forward work-around is to implement a formal profile retrieval (Philipps-Tikhonov 1st order) and to regularize ad infinitum. Then, the algebra just delivers the averaging kernel at the expense of enhanced computational cost since one needs to calculate the layer-wise Jacobians. If computational cost is an issue, one can follow the recipe in section 2.3 of Borsdorff et al., 2014, https://doi.org/10.5194/amt-7-523-2014. Isn't that about the same as what section 5 proposes? Please put your work in relation to the above paper (and consider using their standard notation).

In equ. 21, I cannot follow the last identity. Given that K=dF1(S)/dS (equ 14), why is this equivalent to a derivative involving Delta-y1? Delta-y1 contains F0 i.e. the forward model at the initial state while the Jacobian K needs to be calculated at the iterated state. If the averaging kernel is calculated for the initial state, there might be residual non-linearity errors since for the initial state, it is less likely to be in the "linear neighbourhood" of the true state than for the iterated/final state. These effects are probably small as long as a reasonable prior is chosen and the scaling factor is just about unity. So it is merely a matter of precise understanding, I guess.

[Figure]

3. In the conclusions, Sun et al. highlight that the spectral sampling of the absorption line is superior to simple on/offline lidar setups. Would it be possible to demonstrate/quantify that statement e.g. by mimicking an on/offline measurement by picking two spectral samples?

Technical comments:

L40: uses -> used

L57: for the CO2 is measured -> check sentence structure

L60: seeder -> seed

L80: mereological -> meteorological

L97: check sentence structure

Section 3.3: It would be good to highlight that the scaling factor applies to the below-aircraft CO2 profile, not to the CO2 profile up to top-of-the-atmosphere (as it would for passive solar backscatter techniques since the downward lightpath always travels the entire column). Therefore, the quantity XCO2 is actually the below-aircraft column mole fraction.

L125: into account of the effects -> into account the effects

L144: I wonder whether it would be more straightforward to first define OD via an equation and then define the transmission (equ. 4) via OD.

L161: "as in conventional trace gas sounding" - Does "conventional" refer to passive solar backscatter remote sensing? The passive techniques only formally retrieve profiles that consist of several layers but they effectively regularize the inverse problem such that they end up with $\sim$1 DFS which is equivalent to the total column. The benefit of a formal profile retrieval is that one gains flexibility and the algebra delivers all the diagnostics.

L184: slop -> slope

L186 and following: Is it common (compliant with AMT rules?) to write vectors in capital letters and their elements in regular letters?

L202: Is the measurement error strictly diagonal or is there correlations possible e.g. considering the drift in laser energy?

L204: parameters set -> parameters set \vec{S}

L218: Preconditioning the least-squares with 1/f0 can end up in numerical issues if the transmission approaches zero i.e. if the absorption becomes opaque. While the absorption line used here is not opaque, there might be a word of caution warranted for the reader to consider this aspect in general.

L269: "second row" It should be made clear that it is the second row because the state vector chooses the scaling factor to be the second element. Of course, any ordering is possible in general.

Section 7.3: There is more to inverse estimation than least-squares and (truncated) SVD, for example optimal estimation and Philipps-Tikhonov. As noted above, going to a formal profile retrieval has the benefit, that the algebra delivers the averaging kernels and that one becomes flexible to empirically tune the regularization strength instead of retrieving exactly 1 DFS (total column).

Figure 5: Do I understand correctly that the units for the vertical axis are fractions or is it differences as the caption says? Maybe, one could change the axis label to "measured/modelled" or "measured-modelled", whatever applicable.

Figure 5: Do you think the fit could be improved by using a refined line-shape model e.g. including finer collisional effects or line-mixing or by using a dedicated cross-section database of the narrow wavelength band? I recommend adding a word on these spectroscopic effects in the manuscript.

Figure 7: The surface reflectance of 0.3 looks quite favorable for the experiment. Could you say a word on whether the conditions you encountered are representative of the performance for more extended sets of measurements e.g. over vegetation surfaces.

Figure 7: Could you add a horizontal axis for (approximate) travel distance?

Figure 8: It might be a bit misleading to plot the figure as a height profile since the individual data are representative for the "columns below aircraft" (if I understand correctly). If one wanted to derive the vertical profile, one would need to peel it out from the differences between consecutive data points, right? Maybe, you could guide the reader by putting a caveat into the caption?
* * *

---

## Author Comment (AC2) · 17 Mar 2021

Reviewer 1: General comments This is an interesting manuscript that addresses a novel approach to retrieve the column-integrated mixing ratio of greenhouse gases (XCO2 or XCH4) by measurements with a multi-wavelength pulsed Lidar sounder. XCO2, its variance and other parameters are derived from comparison of the measured transmission line shape function to a modelled one by means of a least squares fit. The presented retrieval method appears to be quite similar to passive sensors that but it is adjusted to serve for the pulsed lidar measurements with its specific averaging kernels. For evaluation purposes, real multi-wavelength Lidar measurements by a specific airborne CO2 Sounder lidar are used to test the proposed algorithm. The retrieved XCO2 values agree pretty well (< 1 ppm) to independent measurements of the mixing ratio column using aircraft in-situ data which is a remarkable result of this study. In a wider context the results of this study demonstrate that active remote sensing in combination to advanced retrieval methods fulfils the stringent measurement requirement on XCO2 for flux inversion experiments as demanded by the top down approach. The overall quality of the manuscript is good and the material presented is very well suited for publication in AMT subject to a few minor questions and comments for improvement of the manuscript as detailed below.

We appreciate your careful review and your many helpful comments. We addressed all of the comments in the revised manuscript, and the responses to the individual comments are given below.

Specific comments Chapter 1 lines 37-46: a brief chapter on the background and motivation for the selected measurement approach and retrieval algorithm is missing here. Different to passive sensors, conventional DAOD-based retrieval techniques don't need a first guess CO2 profile for the calculation of XCO2. What is the reason to give-up this advantage or what is the benefit of the selected approach. Further to this, the measurement precision of active sensors always lacks on sufficient "good" photons, particularly, if applied from space. The highest measurement sensitivity can be achieved at wavelength positions where the DAOD is close to unity (dependent on the background noise). This would imply to spend most of the laser pulses at wavelength positions that are more favorably in terms of measurement sensitivity rather than to emit in the far wing where the DAOD is very small as shown in Fig 2. Could the authors comment or justify the choice of the wavelengths. This question is also related to the discussion chapter 7.2

We added a few more sentences to compare the existing XCO2 retrieval algorithm with DAOD to the multi-wavelength sampling and curve fitting methods described in this manuscript, Lines 40-44: However, these algorithms rely on the accurate knowledge of

the line shape of the $CO_2$ absorption. They are sensitive to measurement biases due uncertainties in spectroscopy and meteorological conditions that affect the line shape. They also require precise knowledge of the laser wavelengths, the lidar receiver optical transmission versus wavelength, and the Doppler shift of the received laser signals." and Lines 49-52: "Although this multiwavelength approach requires more laser power to achieve a given XCO2 measurement precision, it provides more tolerance to the uncertainties in the $CO_2$ absorption line shape, lidar receiver response, and Doppler shift, so that the retrieved XCO2 is more robust against bias errors." Chapter 2: Although the instrument description is not the focus of this issue a table of the most important instrument parameters which largely dominate the measurement performance from instrument point of view would be very beneficial for the manuscript. The missing parameters comprise: the laser pulse energy, the spectral width and spectral purity of the laser pulses, the wavelength domain of the pulse train, the optical filter band width, the laser footprint on ground, the off-nadir sounding angle, and maybe others. This table should also give information on the wavelength stability during a typical averaging period. In particular, the accuracy (stability) of the emitted laser pulses within each pulse train and its reproducibility over the various pulse trains during the measurement. The wavelength stability is a key parameter that impacts on the systematic error.

A table of the instrument parameters is added in Chapter 6, where we show how the algorithm is used with the lidar measurements. We believe it is better place for the table since that is where we discuss the specifics of the instrument.

Chapter 3: Line 115: The transition from Eq: 1 to Eq: 2 is somewhat puzzling. More information on the calibration parameter C2 should be given. It is assumed that the calibration requirement comes from the energy monitor device which is different to the signal detection system in Fig.2. What is calibration procedure of this parameter on ground and what is the stability during airborne measurements.

Eq. (2) has been revised and the instrument constant C2 is replaced by its mathematical expression. We added two sentences, Lines 139-141, to explain how the instrument

optical transmission is calibrated.

Line 130: The simple Eq. 4 assumes monochromatic laser pulses and almost zero emissions elsewhere (high spectral purity), otherwise it would fail. These important instrumental constraints should be added in this paragraph.

We added text at Line 169 to give the laser frequency accuracy (<1 MHz) and linewidth (30 MHz) of the laser transmitter. We also added references about early studies of the effect of laser frequency offsets and noise that showed such laser frequency offset and linewidth are acceptable for XCO2 retrieval compared to other noise sources.

Lines 143-144: The introduction of the layered OD versus column OD is a bit confusing. What is measured and what is modelled? An equation for the measured column OD should be given here.

We give the definition of the column OD and layered OD in Eqs. (6) and (7) to avoid confusion. They are both for the modeled atmosphere, as the entire subsection (3.3) discusses the atmosphere model used to calculate the line shape for the least squares fit.

Line 150: the line shift is also influenced by atmospheric parameters... should be added.

The sentence was revised as " The atmospheric pressure, temperature and water vapor can cause shifts and broadening of the Co2 absorption line which affects the cross sections at measured wavelengths."

Lines 161-163: It is agreed that the information content from measurement in the line wing appears to be rather small because of the small column OD in case of the selected CO2 absorption line. However, passive sensors that are capable to resolve the absorption line profile would face similar problems since a higher spectral resolution results in a lower SNR. This is a confusing side discussion here, therefore it is recommended to drop these lines.

We agree and to avoid this confusion we deleted the comparison to passive measurements. We revised the rest of the text to emphasize that lidar can only make a limited number of measurements across the absorption line with adequate SNR at each sample point due to the limited laser power.

Lines 165-169: Selection of only one scale factor might be a bit too optimistic. It would imply that the selected line parameters and the absorption cross section model are accurate enough to serve for all wavelengths and all atmospheric layers. Could the authors comment on this issue ?

We revised the paragraph to put our retrieval algorithm in context with the techniques used to retrieve column mixing ratio in passive remote sensing. Our approach follows the same method used in the passive remote sensing to retrieve column mixing ratio, but with the modifications needed for lidar's sampling of one absorption line with multiple wavelengths.

Line 210, Eq. 9: further to above, the weighting factors introduced in the loss function are chosen to be similar at all wavelengths, just differing in their SNR. Maybe this is also a too simple approach and need to be justified in the manuscript. It takes not into account the various unfavorable column optical depths of the measurements in the line wing. The more optimal soundings with more optimal column ODs should be given more weight in the retrieval.

The weighting factor matrix in Eq. (9) (now (11) is a generic expression as in all loss function. The SNR squared is the optimal weighting factor at the given wavelength for zero mean random Gaussian measurement noise according to Bevington (1969). We have considered but not thoroughly tested other choices of weighting factors. We did find we need to avoid sampling the line shape at OD >1.5. We added a sentence about this in the Discussion section.

Chapter 5 Line 257: What is the reason present Eq. 19. It is not used in the manuscript

The equation has been deleted as suggested.

Chapter 6 Line 291: What are other possible noise contributions such laser speckles in the Lidar echoes or from pulse energy detection unit which are not considered in Eq. 24.

We added a few sentences in Lines 346-353 to point out that we did not include the effect of laser speckle noise in the expressions because it is usually negligible for our lidar which has a large footprint and which averages a large number of laser pulses per wavelength for each retrieval. We also pointed out that we did not consider the effect of noise in the metrological data used to construct the model of the lineshape.

Page 13 Fig. 5: Several parameters have been fitted in Fig. 5 but only the rms variation of XCO2 is discussed. What are the rms variations of the other fitted parameters and is there any interference to the retrieved XCO2. Further to this, is the quality of the fit robust enough against changes of the first guess profiles CO2 and H2O or against errors of the spectroscopic or meteorological data.

We did not give the rms fitting errors for other variables since the main purpose of the algorithm and Fig. 5 is to retrieve XCO2. We also iterated the XCO2 retrieval several times with the Doppler correction each time based on the solutions from the previous iteration. The error for water vapor is relatively large because of the shallow line depth. The purpose of water vapor retrieval was to avoid bias in the XCO2 retrieval.

Chapter 7.1 As known from many previous studies, systematic measurement errors play a key role in flux inversion experiments, and the requirements are very stringent there. The discussion on this topic is a bit thin. The authors are requested to discuss possible sources of systematic errors which may be related to the processing algorithms (linearization step, weighting factors, scale factor, first guess profiles, and spectroscopy to model the absorption line profile).

The bias in the XCO2 retrieval is indeed an importan topic. Figures 8 shows two cases

where we compared the XCO2 retrieval from airborne lidar measurements to those computed from in situ measurement, which show typical retrieval biases at present using this measurement approach and retrieval algorithm. We are currently studying this topic further through analysis of more in situ comparisons and simulations. The results will be reported in a future publication.

Referee 2 General comments The paper by Sun et al. discusses measurements of an airborne CO2 lidar that operates at multiple wavelengths, thus sampling the atmospheric transmission spectra of a single CO2 line. The paper puts much weight on the description of the least-squares retrieval algorithm for the (below-aircraft) column-average dry-air mole fraction of CO2 (XCO2) and the retrieval diagnostics. The topic is suitable for AMT and the demonstrator results in section 6 show excellent performance. The paper is a bit heavy on the algorithm description (which actually is a standard technique) but this might be justified since the paper aims at making a link from the lidar community to the community that works on passive remote sensing of CO2. Overall, I recommend publication after considering my mostly minor comments below.

We appreciate your careful review and your many helpful comments. We recognize the basic algorithm described here is a standard technique for passive remote sensing. However, it is the first time it is used for retrievals for multiwavelength IPDA lidar measurements. We hope the detailed description of this algorithm here can help to link the two communities.

Comments: 1. The least-squares fit of an atmospheric transmission spectrum is not new. Its detailed description in section 2 through 5 might be justified in the present context where a new experimental technique is further developed. I would see one purpose in connecting the CO2 lidar community to the passive remote sensing community. But, then, the notation chosen by Sun et al. does not really correspond to standard notation e.g. defined in Rodgers, 2000, https://doi.org/10.1142/3171. I find it particularly twisting to use various "overhead" symbols for discriminating between true, a priori and estimated states. While it is not a mandatory request, I would recommend

considering to go closer to standard notation.

Please see our reply above. Also we have changed the variables with overhead bar to subscript "a" for a priori to be consistent with the notation used in Rodgers 2000.

2. I was particularly puzzled by the section on averaging kernels. It is a well-known problem that total column/profile-scaling retrievals complicate the averaging kernel calculation. The most straight-forward work-around is to implement a formal profile retrieval (Philipps-Tikhonov 1st order) and to regularize ad infinitum. Then, the algebra just delivers the averaging kernel at the expense of enhanced computational cost since one needs to calculate the layer-wise Jacobians. If computational cost is an issue, one can follow the recipe in section 2.3 of Borsdorff et al., 2014, https://doi.org/10.5194/amt-7-523-2014. Isn't that about the same as what section 5 proposes? Please put your work in relation to the above paper (and consider using their standard notation).

Thank you for pointing this out. Our averaging kernel is indeed the same as that given in Borsdorff et al. 2014. We have revised the text to use the definition given in that paper without redoing the derivation and gives the expression which is specific to our lidar.

In eq. 21, I cannot follow the last identity. Given that K=dF1(S)/dS (equ 14), why is this equivalent to a derivative involving Delta-y1? Delta-y1 contains F0 i.e., the forward model at the initial state while the Jacobian K needs to be calculated at the iterated state. If the averaging kernel is calculated for the initial state, there might be residual non-linearity errors since for the initial state, it is less likely to be in the "linear neighbourhood" of the true state than for the iterated/final state. These effects are probably small as long as a reasonable prior is chosen and the scaling factor is just about unity. So it is merely a matter of precise understanding, I guess.

We are grateful for your careful review and critique of our description of the averaging kernel and for the reference, We have now used the standard definition of the column averaging kernel given in Borsdorff et al., 2014, which eliminates the need for the

derivation of the averaging kernel and Eq. 21. We also combined Section 6 with Section 5. These changes standardized our retrieval and simplified the paper.

3. In the conclusions, Sun et al. highlight that the spectral sampling of the absorption line is superior to simple on/offline lidar setups. Would it be possible to demonstrate/quantify that statement, e.g. by mimicking an on/offline measurement by picking two spectral samples?

We are currently working on a computer simulation of XCO2 retrievals using the algorithm described in this paper. We plan to study the effect of the number of laser wavelengths and their placement across the CO2 absorption line, including special cases with just the on/offline measurements. The results will be reported in a future publication. We also added a short paragraph in the Discussion section that the algorithm described in this paper can be used in on/offline IPDA lidar (Line 434).

Technical comments: Thanks for pointing out these typos and grammatical errors.

L40: uses -> used Done.

L57: for the CO2 is measured -> check sentence structure The sentences have been revised.

L60: seeder -> seed Done.

L80: mereological -> meteorological Done.

L97: check sentence structure The sentence has been revised.

Section 3.3: It would be good to highlight that the scaling factor applies to the below aircraft CO2 profile, not to the CO2 profile up to top-of-the-atmosphere (as it would for passive solar backscatter techniques since the downward light path always travels the entire column). Therefore, the quantity XCO2 is actually the below-aircraft column mole fraction.

We added a sentence: "Note that Eq. (3) and the XCO2 measurement is for the

atmosphere column from the lidar to the surface. This is different from the passive remote sensing measurement where the incident light from the sun and the reflected light are at an angle and go through different atmosphere columns."

L125: into account of the effects -> into account the effects

Done.

L144: I wonder whether it would be more straightforward to first define OD via an equation and then define the transmission (eq. 4) via OD.

We feel it is more intuitive, at least for the lidar community, to start with the atmospheric transmission since it is directly measured by the lidar.

L161: "as in conventional trace gas sounding" - Does "conventional" refer to passive solar backscatter remote sensing? The passive techniques only formally retrieve pro files that consist of several layers but they effectively regularize the inverse problem such that they end up with âĹ‘ij1 DFS which is equivalent to the total column. The benefit of a formal profile retrieval is that one gains flexibility and the algebra delivers all the diagnostics.

We deleted the sentence that seems to imply that passive remote sensing can readily retrieve the profile. The text has been revised as: "... Although Ramanathan et al. (2018) showed a few additional parameters about the $CO_2$ absorption line shape may be retrieved, they provide only limited information about the vertical profile of $CO_2$ mixing ratio. Therefore, we choose to retrieve a single scale factor for a reference profile, similar to the profile-scaling used in passive remote sensing (Borsdorff et al. 2014). ..."

L184: slop -> slope

Done.

L186 and following: Is it common (compliant with AMT rules?) to write vectors in capital

letters and their elements in regular letters? We find this is the case at least in some AMT articles, e.g., Borsdorff et al., 2014. Vectors can also be expressed in boldface regular (lower case) letters.

L202: Is the measurement error strictly diagonal or is there correlations possible e.g. considering the drift in laser energy?

The lidar measurement errors at different wavelength are mutually uncorrelated. The dominant noise sources are the photodetector shot-noise and the thermal noise from the detector preamplifier which are independent on each measurement. The transmitted laser pulse energies at the different wavelengths are also independently measured and are corrected for in the lidar measurements used in the retreival.

L204: parameters set -> parameters set \vec{S} Done.

L218: Preconditioning the least-squares with 1/f0 can end up in numerical issues if the transmission approaches zero, i.e. if the absorption becomes opaque. While the absorption line used here is not opaque, there might be a word of caution warranted for the reader to consider this aspect in general. We added a short paragraph after Eq. (15): "The use of the above normalization greatly simplifies the mathematical derivation as well as the data processing since it cancels out the exponential terms in \mathbit{F}\left(\mathbit{S}\right). However, this technique can only be used when the values of the forward model f0\left(\lambda_i\right) are not approaching zero at all sampling wavelengths."

L269: "second row" It should be made clear that it is the second row because the state vector chooses the scaling factor to be the second element. Of course, any ordering is possible in general.

The sentence has been revised to "...the row of the G matrix for calculating the XCO2 scale factor ..."

Section 7.3: There is more to inverse estimation than least-squares and (truncated)

SVD, for example optimal estimation and Philipps-Tikhonov. As noted above, going to a formal profile retrieval has the benefit, that the algebra delivers the averaging kernels and that one becomes flexible to empirically tune the regularization strength instead of retrieving exactly 1 DFS (total column).

The objective of this paper is to report how we adapted the least squares fit method from the passive remote sensing community for retrievals of column XCO2 from multi-wavelength lidar measurements. We recognize there are more advanced retrieval algorithms that can take advantage of prior knowledge of the vertical profile and achieve better results. They are beyond the scope of this paper. We will look into these retrieval algorithms, such as the Philipps-Tikhonov regularization, in future studies.

Figure 5: Do I understand correctly that the units for the vertical axis are fractions or is it differences as the caption says? Maybe, one could change the axis label to "measured/modelled" or "measured-modelled", whatever applicable.

The axis label for the lower plot in Figure 5 has been changed to "measured/modeled."

Figure 5: Do you think the fit could be improved by using a refined line-shape model e.g. including finer collisional effects or line-mixing or by using a dedicated cross-section database of the narrow wavelength band? I recommend adding a word on these spectroscopic effects in the manuscript.

We are conducting simulations to investigate if the fit can be improved with refined line shape, and hopefully can report our findings at a later time.

Figure 7: The surface reflectance of 0.3 looks quite favorable for the experiment. Could you say a word on whether the conditions you encountered are representative of the performance for more extended sets of measurements e.g. over vegetation surfaces. We added a sentence: "The ground surface over this stretch of the flight was dry desert and the sky is clear at the time."

Figure 7: Could you add a horizontal axis for (approximate) travel distance?

We updated the figure caption that the distance for the segment of the flight was about 160 km at an approximate airplane speed of 200 m/s.

Figure 8: It might be a bit misleading to plot the figure as a height profile since the individual data are representative for the "columns below aircraft" (if I understand correctly). If one wanted to derive the vertical profile, one would need to peel it out from the differences between consecutive data points, right? Maybe, you could guide the reader by putting a caveat into the caption?

We added in the caption: "The column XCO2 from the aircraft to the surface retrieved from the lidar measurements . . ."